# The HCN domain couples voltage gating and cAMP response in hyperpolarization-activated cyclic nucleotide-gated channels

Alessandro Porro[1], Andrea Saponaro[1], Federica Gasparri[1], Daniel Bauer[2], Christine Gross[2], Matteo Pisoni[1], Gerardo Abbandonato[1], Kay Hamacher[2], Bina Santoro[3], Gerhard Thiel[2], Anna Moroni[1]*

[1]Department of Biosciences, University of Milan, Milan, Italy; [2]Department of Biology, TU-Darmstadt, Darmstadt, Germany; [3]Department of Neuroscience, Columbia University, New York, United States

**Abstract** Hyperpolarization-activated cyclic nucleotide-gated (HCN) channels control spontaneous electrical activity in heart and brain. Binding of cAMP to the cyclic nucleotide-binding domain (CNBD) facilitates channel opening by relieving a tonic inhibition exerted by the CNBD. Despite high resolution structures of the HCN1 channel in the cAMP bound and unbound states, the structural mechanism coupling ligand binding to channel gating is unknown. Here we show that the recently identified helical HCN-domain (HCND) mechanically couples the CNBD and channel voltage sensing domain (VSD), possibly acting as a sliding crank that converts the planar rotational movement of the CNBD into a rotational upward displacement of the VSD. This mode of operation and its impact on channel gating are confirmed by computational and experimental data showing that disruption of critical contacts between the three domains affects cAMP- and voltage-dependent gating in three HCN isoforms.

*For correspondence:
anna.moroni@unimi.it

Competing interests: The authors declare that no competing interests exist.

## Introduction

Hyperpolarization-activated cyclic nucleotide-gated (HCN) channels (*Gauss et al., 1998*; *Ludwig et al., 1998*; *Santoro et al., 1997*; *Santoro et al., 1998*) are the molecular determinants of the If/Ih current, a mixed $Na^+/K^+$ current that controls pacemaking in cardiac and neuronal cells (*DiFrancesco, 1993*; *Pape, 1996*). Unique among voltage-gated channels, HCN channels are activated by hyperpolarization of membrane voltage. Upon activation, their inwardly directed cation current slowly depolarizes pacemaker cells to the threshold for action potential firing. In addition to voltage, HCN channels are modulated by the second messenger cAMP, which enhances channel open probability, thus shifting the voltage-dependency of opening to more positive values and increasing the amount of current at any given voltage (*Wainger et al., 2001*). Regulation by cAMP of native HCN channels plays crucial physiological roles: in the sinoatrial node, it underlies the autonomic regulation of the heartbeat (*DiFrancesco and Tortora, 1991*); in pathophysiological conditions, such as in peripheral neuropathic pain, cAMP modulation of HCN channels leads to neuronal hyperexcitability enhancing pain transmission (*Emery et al., 2011*). Despite the obvious physiological relevance and possible therapeutic applications, dual activation of HCN channels by voltage and cAMP is still poorly understood at both structural and functional levels.

The recently solved structure of HCN1 (*Lee and MacKinnon, 2017*) substantially advanced our comprehension of HCN channel architecture. Each subunit of the tetrameric channel is formed by six (S1-S6) transmembrane domains (TM). Helices S1-S4 form the voltage sensor domain (VSD), which is connected to the pore domain (helices S5-S6). The C terminal cytosolic domain, attached to the end of TM S6, includes the C-linker (helices A' and B' connected via a loop to helices C' and D') and the

cyclic nucleotide-binding domain (CNBD), a beta strand-alpha helix fold that hosts the cAMP binding pocket (*Zagotta et al., 2003*). The HCN1 structure further revealed that a conserved sequence at the cytosolic N terminus, named HCN domain (HCND), folds into three short helices (HCNa, HCNb and HCNc) that wedge in between the VSD and C-linker (*Lee and MacKinnon, 2017*). Thus, the tetrameric HCN1 channel shows a modular architecture with the two regulatory domains, the VSD and the C-linker/CNBD, respectively, connected to either side of the pore and seemingly bridged by the HCN domain.

The membrane-embedded VSD is connected to the N terminus of the pore via the S4-S5 linker. The VSD moves within the electrical field upon hyperpolarization (*Bell et al., 2004*; *Männikkö et al., 2002*; *Vemana et al., 2004*) but it is not yet clear how this movement is coupled to pore opening. It has been suggested that, unlike in voltage-gated K$_v$ channels, the S4-S5 linker is not required for hyperpolarization-dependent activation in HCN channels. Indeed, in the closely related SpHCN channel, the S4-S5 linker can be cut without compromising voltage gating (*Flynn and Zagotta, 2018*). A potential mechanism of coupling between the VSD and the pore has been proposed based on the cryo-EM structure of HCN1 (*Lee and MacKinnon, 2017*). This structure reveals a parallel arrangement of TM S4 and S5, comprising multiple contacts between these domains, which may serve to directly transmit the VSD movement to the pore without the need for an involvement of the S4-S5 linker (*Cowgill et al., 2019*).

At the other end, the CNBD is connected to the C terminus of the pore via the C-linker. The four C-linkers are a continuation of the S6 pore helices and tetramerize to form a disc-like structure, called the 'gating ring'. Binding of cAMP to the CNBD induces an iris-like rotation in the gating ring, which is thought to help unwrap the bundle of S6 helices and open the pore (*Gross et al., 2018*; *Lee and MacKinnon, 2017*; *Marchesi et al., 2018*; *Shin et al., 2004*; *Weißgraeber et al., 2017*).

The two regulatory signals, voltage and cAMP, affect the channel open probability in a hierarchical manner. Voltage is both necessary and sufficient to open the channel, as HCN1 mutants lacking the cytosolic C-linker/CNBD domain are fully regulated by voltage (*Wainger et al., 2001*). This indicates that the VSD does not require the rotation of the C-linker gating ring in order to open the pore, but presumably acts via a membrane-delimited pathway. Conversely, cAMP alone is not sufficient to open the channel. In the absence of hyperpolarizing voltages, the channel remains closed even in the presence of high concentrations of cAMP (*Gauss et al., 1998*; *Ludwig et al., 1998*; *Santoro et al., 1998*). It is therefore not surprising that in the HCN1 structure, which was obtained at 0 mV, the conformation of the pore is closed even in the presence of cAMP. It is striking, however, that the structure of the gating ring is basically identical in the cAMP-bound (holo) and unbound (apo) HCN1 structures. Thus, in the presence of cAMP, the gating ring only marginally rotates by about 1 Å and the displacement of S6 is correspondingly minimal (*Lee and MacKinnon, 2017*). These functional and structural observations suggest that the gating ring movement must be under the control of voltage, implying the existence of a direct physical contact between the VSD and the C-linker.

A direct connection between the C-linker/CNBD and the VSD is also strongly suggested by previous studies in which ligand binding and channel activation were simultaneously measured. Thus, *Kusch et al. (2010)* demonstrated that hyperpolarizing voltages enhance the affinity of the HCN2 channel for cAMP by up to 3-fold. In the same study, Kusch et al further showed that the increase in affinity precedes channel activation, underscoring a direct connection between the C-linker/CNBD and VSD, which can occur in either direction, and bypassing the pore. It is worth noting that this allosteric control of VSD on the affinity of CNBD for cAMP most likely accounts for the difference in affinity measured for the isolated CNBD (*Lolicato et al., 2011*; *Saponaro, 2018a*; *Saponaro et al., 2018b*; *Saponaro et al., 2014*) with respect to those measured in the full length HCN protein (*Kusch et al., 2010*; *Thon et al., 2015*).

More recently, the same authors reported that binding of cAMP to the CNBD affects the gating charge of the voltage-dependent opening (*Hummert et al., 2018*), which provides yet another proof for the model described above.

Despite the evidence laid out above, a direct physical interaction between the two signal sensing domains, the VSD and the C-linker/CNBD, has not been described so far. A close inspection of the HCN1 structure reveals that the HCND appears to contact both the VSD and the C-linker, suggesting the possibility that the VSD and the CNBD may communicate *via* the HCND without a requirement for the pore as a connecting element. Therefore, in the present study, we set out to test the

hypothesis that the HCND serves as a 'trait d'union' between the two regulatory domains. We provide evidence that the HCND is indeed critical for the cAMP response in all HCN isoform tested (HCN1, HCN2 and HCN4). In addition, we uncover a second important role of the HCND in gating, which is to set the range of voltages at which the channel opens, thus ensuring channel operation at physiological relevant voltages only.

## Results

The HCND, shown in orange in *Figure 1A*, inserts in a wedge-like manner between the VSD and the C-linker gating ring, respectively shown in violet and green. It folds into three short alpha helices HCNa, HCNb and HCNc (*Figure 1B*). HCNa lies parallel to the plane of the membrane, HCNb faces the C-linker ring, and HCNc ends in a loop which penetrates the membrane plane and connects the HCND to S1, the first TM helix of the VSD. By inspecting the HCN1 structural model and density map, we found that the HCND appears to directly contact both the VSD and the C-linker. A zoom into the region of interest highlights that the HCND is anchored to the VSD of its own subunit by

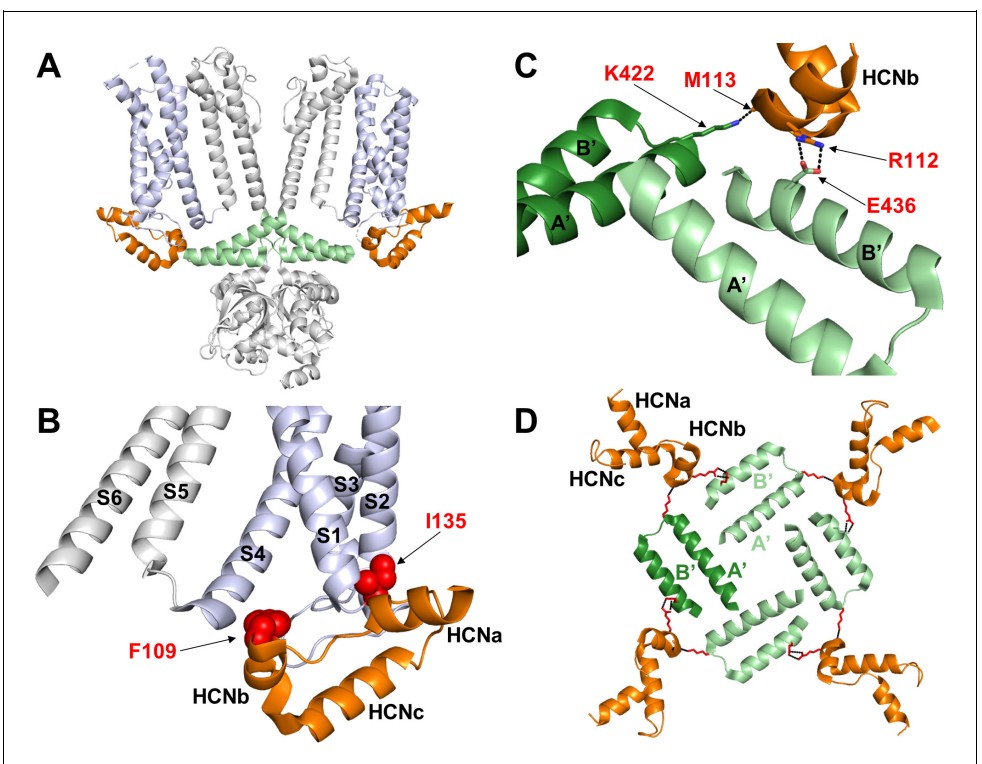

**Figure 1.** Hydrophobic and hydrophilic interactions established by the HCN domain with the voltage sensor domain and the C-linker. (A) Side view of two opposite subunits of HCN1 (PDB_ID: 5U6O) with the HCN domain (HCND) shown in orange, the voltage sensor domain VSD (TM S1-S4) in violet and C-linker 'elbow' (helices A' and B') in green. The pore domain (TM S5-S6), the CNBD and helices C'-D'-E'-F' of the C-linker are in grey. (B) Blow-up of the structure in (A) showing the position of residues F109 and I135 of the HCND engaged in hydrophobic interactions with the VSD of the same subunit. Red spheres represent the van der Waals surface of the side chains of F109 and I135. (C) Detailed view of the electrostatic interactions between the HCND and the C-linker helices belonging to the adjacent and opposite subunit (light green, adjacent; dark green, opposite). R112 and M113 of HCND interact, respectively, with E436 and K4226 of the C-linker. Salt bridges are represented as dashed black lines. Residues of interest are labelled in red using hHCN1 numbering. HCNa,b and c are labelled according to *Lee and MacKinnon (2017)*. (D) Top view of the tetrameric arrangement of the C-linker helices A' and B' showing their interactions with the HCND. The side chains of the residues engaged in the electrostatic interactions connecting the HCNDs to the C-linker ring are shown as red sticks. A' and B' helices of the C-linker belonging to the adjacent or opposite subunit (referring to the labelled HCN domain on the top left only) are colored in light and dark green, respectively.

two hydrophobic residues, F109 and I135, that insert their side chains into two pockets either side of S1 (*Figure 1B*). The same HCND furthermore establishes hydrophylic contacts with the C-linkers of two other subunits. *Figure 1C* shows two residues in the HCNb helix, R112 and M113, which form salt bridges with E436 on the B' helix of the adjacent subunit (light green) and with K422 on the A' helix of the opposing subunit (dark green), respectively. *Figure 1D* provides a top view of the tetrameric assembly of the C-linker helices A' and B' in the tetramer. Each C-linker is contacted by two different HCNDs and each HCND contacts the C-linkers of two different subunits. This structural arrangement leads to the prediction that C-linker movements might be directly transmitted to the VSDs and vice-versa.

To test this hypothesis, we systematically disrupted each of the contacts seen between the HCND and the VSD or C-linker through site-directed mutagenesis and analyzed the properties of the mutant channels in terms of voltage-dependency and cAMP regulation. Given the limited response of HCN1 to cAMP, the mutations were initially introduced in the background of HCN2, an isoform that responds to cAMP with a larger shift (12–15 mV) of the half-activation voltage ($V_{1/2}$) value. Key mutations were subsequently further tested on the background of HCN1 and HCN4 (note that all residues examined are conserved among the isoforms studied here). Residue numbering hereafter refers, if not otherwise specified, to mouse HCN2.

## First hydrophobic pocket

The first residue we analyzed is that of I177 (I135 in HCN1) inserted in a hydrophobic pocket formed by the N-terminal and the C-terminal ends of S1 and S2, respectively (*Figure 2A*). Specifically, residues F182 of S1 and L234 of S2 contact the side chain of I177, located in the unstructured loop that connects HCNc to S1. The interaction of I177 with the VSD is further stabilized by the preceding isoleucine, I176, whose side chain runs parallel to the membrane plane pointing towards a cytosolic loop (*Figure 2B*). This loop, which connects TM S2 to TM S3 of the VSD, in turn contains a stretch of hydrophobic residues, I248, I249 and L250 which presumably interact with the I176 side chain.

We substituted I177 either with a charged aspartic acid (I177D) to perturb the hydrophobic interaction between the HCND and the VSD or with a glycine (I177G) to remove the side chain. The mutant channels were co-transfected with green fluorescent protein (EGFP) in HEK293T cells and patch-clamp experiments were performed after 24 or 48 hr from transfection on EGFP positive cells. We did not detect any HCN-like current from either of the two mutants (I177D, n = 13; and I177G, n = 18) (*Figure 2C*). Confocal microscopy analysis indeed demonstrated that an EGFP tagged version of the respective channels accumulates in intracellular membranes and does not reach the plasma membrane (*Figure 2—figure supplement 1*). In contrast, substituting I177 with valine or alanine to gradually reduce the length of the hydrophobic side chain generated currents indistinguishable from the wild type (wt) channel in terms of amplitude and gating properties (*Figure 2C,D* and *Supplementary file 1A*), suggesting that a hydrophobic side chain of any length in position 177 is required for proper channel folding and/or trafficking to the plasma membrane.

To test if this is a general requirement for HCN channels, key substitutions in the I177 residue of HCN2 were further tested in the background of HCN1. Similar to HCN2, HCN1 channels carrying the I135V/A mutation (corresponding to I177V/A in HCN2) generated a current that was indistinguishable from wt (I135A current shown in *Figure 2—figure supplement 2* and *Supplementary file 1B*; I135V, data not shown), while the I135G HCN1 mutant generated no detectable current (*Figure 2—figure supplement 2* and *Supplementary file 1B*). Confocal imaging of HEK293T cells expressing an EGFP tagged HCN1 channel showed that the I135G mutant also does not reach the plasma membrane (*Figure 2—figure supplement 3*). This confirms that a hydrophobic side chain in this position is, also in HCN1, crucial for correct folding and/or trafficking of the channel to the plasma membrane.

Next, we tested the role of the preceding isoleucine, I176 in HCN2 (I134 in HCN1), which in the HCN1 structure is pointing towards the hydrophobic stretch in the S2-S3 loop (*Figure 2B*). Mutation into aspartic acid, I176D, resulted in no current and no channel at the plasma membrane (n = 13, *Figure 2C*, *Figure 2—figure supplement 1* and *Supplementary file 1A*) indicating that this position too does not tolerate a charged residue. When the isoleucine was substituted with a valine (HCN2 I176V), the channel generated a current indistinguishable from wt (*Figure 2C,D* and *Supplementary file 1A*). When the length of the side chain was decreased with the I176A mutation, we recorded a functional HCN-like current in 28% of cells expressing HCN2 I176A (n = 10 out

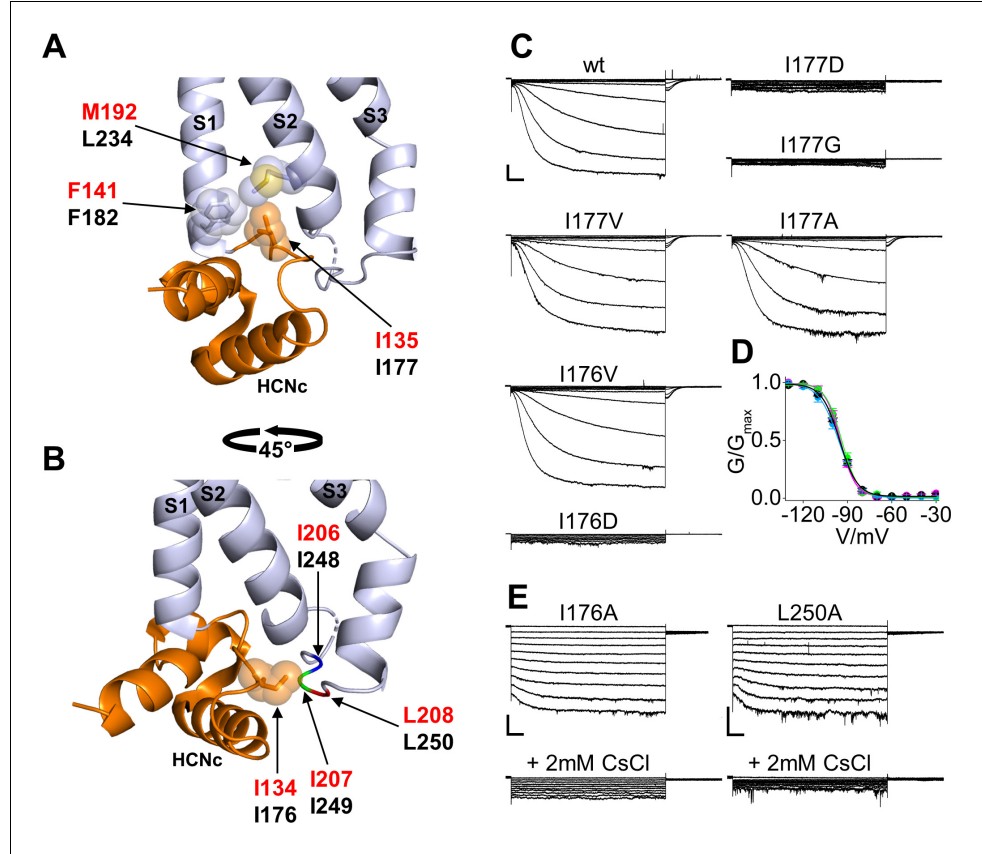

**Figure 2.** Interaction of HCND with the voltage sensor domain: mutational analysis in the first hydrophobic pocket. (A) Detailed view of one HCN1 subunit (PDB_ID: 5U6O) color coded as in *Figure 1* showing I135 side chain inserted in the hydrophobic pocket formed by F141 in the TM S1 and M192 in the TM S2 of the VSD. The side chains of the residues involved in the hydrophobic interactions are shown as sticks surrounded by spheres representing the occupied van der Waals surface. (B) Detailed view of I134 pointing towards the cytosolic S2-S3 loop. The backbones of hydrophobic residues I206, I207, and L208 of the loop are shown in different colors; their side chains are not assigned in the structure (PDB_ID: 5U6O). Red labels: hHCN1 numbering; black labels: mHCN2 numbering. (C) Representative whole-cell currents of wt HCN2 channels and I177D/G/V/A and I176V/D mutants. The voltage step protocol is described in Materials and methods. (D) Activation curves (mean values ± SEM) from wt (black), I177V (green), I177A (cyan) and I176V (magenta) mutant channels. Lines show data fit to a Boltzmann function (*Equation (1)* in Materials and methods). Calculated half activation potential ($V_{1/2}$) and inverse slope factor ($k$) values together with details on statistical analysis are reported in *Supplementary file 1A*. (E) Representative whole-cell current of I176A or L250A mutant channels, before (top) and after (bottom) the addition of 2 mM CsCl to the external solution. Scale bars are 100 pA x 500 ms.
The online version of this article includes the following figure supplement(s) for figure 2:

**Figure supplement 1.** Cellular localization of EGFP-HCN2 I177D/G and I176D mutant channels.
**Figure supplement 2.** Analysis of currents recorded from HCN1 I134, I135 and F109 mutants.
**Figure supplement 3.** Cellular localization of EGFP-HCN1 I135G, I134A and F109E mutant channels.

---

of 35 cells tested). In these cells, we observed a drastic loss of the time-dependent component accompanied by an increase in the instantaneous component of the current (*Figure 2E*). Application of extracellular cesium (2 mM CsCl), a known HCN blocker, inhibited the HCN2 I176A current almost completely, confirming that the instantaneous component is indeed generated by the mutant channel (*Figure 2E*).

Interestingly, introducing an alanine mutation in L250, a residue of the S2-S3 loop facing I176, produced the same phenotype of I176A, including a decrease in the expression levels (n = 6 out of 27, or 22% of tested cells exhibited a measurable HCN current) and a loss of the time dependent component accompanied by an increase of a cesium-sensitive instantaneous component

(*Figure 2E*). These findings confirm the interaction of I176 with the S2-S3 loop that we observed in the HCN1 structure.

Equivalent mutations introduced in HCN1 confirmed that the hydrophobic residue in position 134 (176 in HCN2) is also required for correct channel folding and/or trafficking of this isoform. Mutation I134V resulted in wt-like current. However, unlike I176 in HCN2, residue I134 in HCN1 did not tolerate even the mild substitution I134A. Thus, HCN1 I134A resulted in no currents recorded (*Figure 2— figure supplement 2B*) and in a lack of channel localization at the plasma membrane (*Figure 2—figure supplement 3*).

Taken together these results suggest that a hydrophobic interaction between select residues in the HCN domain and the VSD is essential for proper channel folding, trafficking and function. Even a mild disruption of the interaction between the HCND and the S2-S3 loop, as postulated in the HCN2 I176A and L250A mutants, can result in a significant loss of the voltage-dependent control on channel gating. Whether I176A/L250A act on the function of the VSD directly via their HCND/S2-S3 loop contact, or indirectly by disrupting the HCND/S1-S2 contacts of the neighboring I177 residue, remains unclear.

## Second hydrophobic pocket

The HCND is further anchored to the VSD by means of another hydrophobic residue, a phenylalanine (F109 in HCN1, F151 in HCN2), found in the HCNb helix. In the HCN1 monomer, the aromatic side chain of this phenylalanine is inserted between helices S1 and S4 in a second hydrophobic pocket formed by I284 and M287 in S4, and Y138 in the HCNc-S1 loop (*Figure 3A*).

Computational data support the hypothesis of a strong and stable hydrophobic interaction between the HCND and the VSD with the side chain of F109 inserted into the S1-S4 pocket. A 50 ns long molecular dynamics simulation (MDs) of HCN1 shows F109 stably inserted in this hydrophobic pocket (*Figure 3B*). MDs was repeated with five F109 substitution mutants: tryptophan, methionine, valine, and alanine to reduce stepwise the length of the hydrophobic side chains, and the anionic residue glutamate to disrupt hydrophobic interactions. F109E was the only case in which the side chain was found to rotate out of the pocket (*Figure 3B*) and interact with the cytosolic solvent.

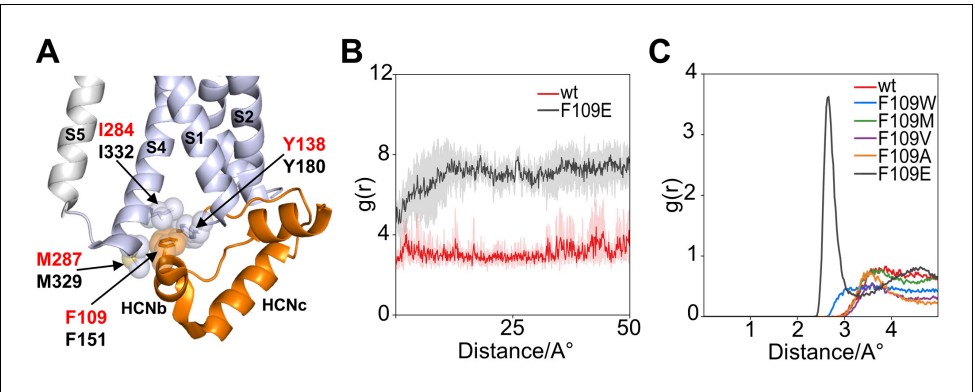

**Figure 3.** Interaction of HCND with the voltage sensor domain: MD simulation on the second hydrophobic pocket. (**A**) Enlarged view of HCN1 subunit (PDB_ID: 5U6O) color coded as in *Figure 1* showing the hydrophobic interactions of F109 with I284 and M287 from TM S4 and Y138 from the loop connecting HCND to TM S1. Spheres represent the van der Waals surface occupied by the side chains. Helices S1 and S4 of the VSD, S5 of the pore domain, HCNb and HCNc of the HCND are labelled. Residues are labelled either in red or black using hHCN1 or mHCN2 numbering, respectively. (**B**) Shortest measured distances between side chain atoms of residue 109 and Y138 in wt (red) and F109E (black) channels over simulation time (50ns). For each simulation time step, the distance between all atoms of then respective side chains were measured and the shortest found distance was plotted over time. Solid lines indicate the average over the four subunits with translucent error bands in the back. It can be seen that the distance for F109E increases during the first 10ns of simulation because the side chain rotates out of the hydrophobic pocket. In contrast, the wt side chain stays inserted in the pocket and close to residue 138. (**C**) Radial distribution function g(r) for water oxygen atoms around the side chain of residue 109. The g(r) plot describes the probability of finding a water molecule at a given distance from the side chain. Only F109E is fully solvated and shows a peak for the first solvation shell of the γ caboxyl group.

*Figure 3C* shows Solvent Radial Distribution Analysis for all residues. The g(r) variable, plotted as a function of distance, describes the probability of finding a water molecule at a given distance from the side chain. As a matter of fact, F109E was the only case where a distance of less than 2 Å from the water molecules could be measured, indicating that the side chain is hydrated. This result supports the idea that F109E prevents cAMP effect because it does not contact the VSD via the second hydrophobic pocket.

Based on this information, we mutated F151 in HCN2, equivalent to F109 in HCN1, into the above five aminoacids. All channels produced HCN-like currents from which we could derive voltage-dependent activation curves (*Figure 4A* and *Figure 4—figure supplement 1*). Decreasing the length of the hydrophobic side chain progressively shifted the half-activation voltage parameter $V_{1/2}$ to more depolarized voltages (*Figure 4A,B* and *Figure 4—figure supplement 1A*). This right shift was $4.4 \pm 0.6$ mV for F151V and $13 \pm 0.4$ mV for F151A. A major shift of $34.7 \pm 0.9$ mV was observed for F151E, a mutation that in the MD simulation of HCN1 prevented side chain insertion in the pocket (*Figure 3B*). Thus, the experimental data are in good agreement with the simulation results, in that a weakening of the hydrophobic interaction is paralleled by a right shift of the voltage activation curve. Disruption of the pocket presumably also affected channel folding and/or trafficking to the plasma membrane, as currents could be measured only in 6 out of 31 cells tested (~19%) for the more extreme HCN2 F151E mutant (*Supplementary file 1A*).

Overall, the combination of results from experimental and computational data is consistent with the prediction that the HCND interacts with the VSD through critical hydrophobic contacts established by residue F109/F151 in HCN1/HCN2, respectively. This interaction has an impact on the VSD regulation of the channels' activation curve, as a progressive weakening of the interaction established by F151 concomitantly shifts the $V_{1/2}$ to more depolarized voltages.

The finding that the HCND controls the VSD makes the testable prediction that it may also control the response of the VSD to cAMP. We therefore tested the effect of cAMP on the F151 mutants described above. Mutants F151W/M/V/A responded to cAMP like the wt, with a ~ 15 mV positive shift of their $V_{1/2}$ value (*Figure 4*, *Figure 4—figure supplement 1* and *Supplementary file 1A*). This suggests that the sensitivity of the channel to cAMP is not compromised as long as the hydrophobic interaction is maintained to some degree. In contrast, the F151E mutant, in which such interaction is fully disrupted, is also completely insensitive to cAMP (*Figure 4* and *Figure 4—figure supplement 1A*). This is consistent with the hypothesis that the interaction between the HCND and the VSD is required for the allosteric regulation of voltage gating by cAMP.

We tested whether other charged substitutions (D, K, R) have a similar effect. In two cases, F151D and F151K, we couldn't measure a current (0/21, 0/26 cells), while F151R expressed a current indistinguishable from wt in terms of $V_{1/2}$ and cAMP-response, in 10/24 cells (*Figure 4—figure supplement 2A,B*). Molecular dynamics simulation confirmed that F151D and F151K are exposed to the solvent while F151R is found predominantly in the hydrophobic pocket where it forms an hydrophilic interaction with D183, a residue on S1. This interesting behaviour of arginine can depend on the hydration properties of its guanidinium moiety that makes dehydration energetically less costly than that of the aliphatic amino group of lysine and to the length of the side chain that can sample conformational space extensively (*Armstrong et al., 2016*; *Harms et al., 2011*).

Equivalent mutations to alanine and glutamate were next introduced in residue F109 in HCN1. However, unlike F151 in HCN2, residue F109 in HCN1 did not tolerate even the mild substitution F109A. Thus, HCN1 F109A currents were rarely measurable (6 out of 20 cells, or 30%) indicating poor folding and/or localization of the channel at the plasma membrane. Cells with measurable currents displayed an average right shift in the activation curve of 7 mV (*Figure 2—figure supplement 2* and *Supplementary file 1B*), matching the phenotype of the equivalent mutant F151A in HCN2. Mutation HCN1 F109E was not measurable and imaging experiments confirmed that the channel was retained in internal membranes (*Figure 2—figure supplement 3*).

## C-linker contacts

Inspection of the HCN1 structure reveals that two nearby residues on the HCNb helix, R112 and M113, can form salt bridges with residues K422 on helix A' and E436 on helix B' of the C-linkers of two different subunits (*Figure 1*). Through these contacts, the HCND would be directly connected to the C-linker and could, in principle, transmit its movement to the VSD and vice-versa, the movement of the VSD to the C-linker.

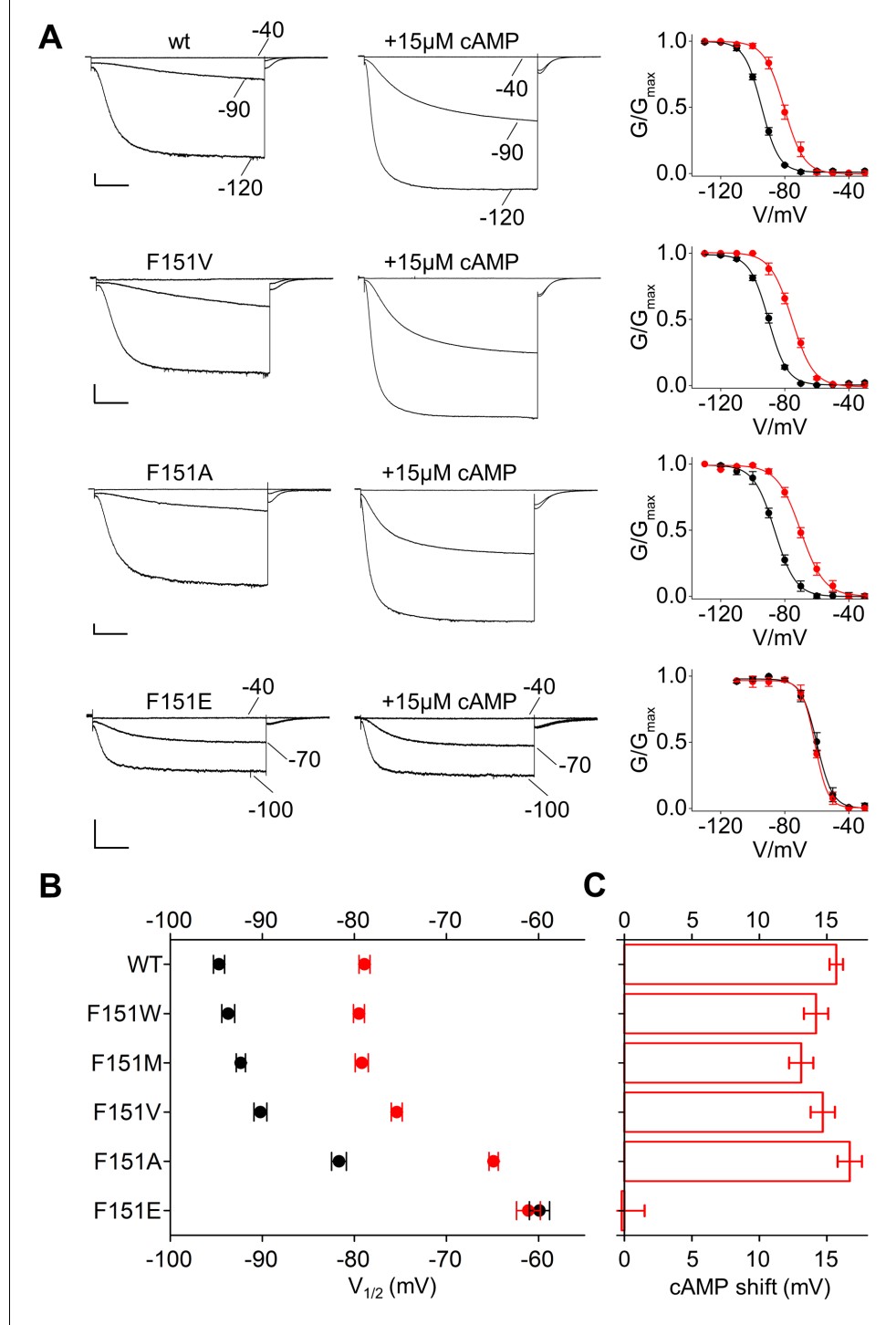

**Figure 4.** Interaction of HCND with the voltage sensor domain: mutational analysis of HCN2 residue F151. (**A**) Representative whole-cell currents recorded, at the indicated voltages, from HCN2 wt, F151V, F151A and F151E channels in the absence and in the presence of 15 µM cAMP. Graphs to the right show corresponding mean activation curves, without (black) and with (red) cAMP. Lines show data fit to a Boltzmann function (Materials and methods) providing half activation potential ($V_{1/2}$) and inverse slope factor (k) values reported in *Supplementary file 1A*. (**B**) Mean $V_{1/2}$ values of all F151 mutants, including F151M/W (currents and activation curves shown in *Figure 4—figure supplement 1*). Black symbols, control; red symbols, + 15 µM cAMP. (**C**) cAMP-induced shift in $V_{1/2}$. Values measured for F151W/M/V/A are not significantly different from wt. Data are shown as
*Figure 4 continued on next page*

*Figure 4 continued*
mean ± SEM. Scale bar: 200 pA x 500 ms. $V_{1/2}$ values, slope factors and statistical analysis are reported in
***Supplementary file 1A***.
The online version of this article includes the following figure supplement(s) for figure 4:

**Figure supplement 1.** Analysis of currents recorded from HCN2 F151 mutants.
**Figure supplement 2.** Patch-clamp analysis and MD simulation of HCN2 F151R, F151K and F151D channels.

To test the predicted mechanical continuum within HCN1 protein, we used linear response theory (LRT), introduced by ***Ikeguchi et al. (2005)***. This mechanical model calculates the direction of conformational changes in a protein upon external perturbation. Therefore, the protein is reduced to a network of beads and springs (anisotropic network model) and the external perturbation, for example binding of a ligand, is mimicked by an external force. Compared to molecular dynamics simulations, this coarse-grained technique requires much shorter computational time and allows insights into conformational changes on larger timescales. In our previous study (***Gross et al., 2018***) we have already successfully simulated binding of cAMP to the CNBD in HCN1 by applying an external force on the 'elbow' of the C-linker. Here we used the HCN1 anisotropic network model to test if C-linker is mechanically connected to HCND and VSD. The computational data show that a force displacement of the C-linker (***Figure 5A***, position 425), which mimics the conformational changes induced by cAMP binding (***Gross et al., 2018***), causes a concerted movement of the C-linker and the HCND; both domains rotate with a similar angle to the z-axis (***Figure 5B***). To examine the consequence of this concerted movement, we analyzed its impact on the VSD. The data in ***Figure 5C*** show that the rotational movement of the C-linker and the HCND is indeed transmitted to the VSD, and to the S4 helix in particular, which tilts in response to the movement of the other two domains (***Figure 5C***). The concerted movement and the transmission of rotational movement of the C-linker and HCND into a tilting movement of S4 is illustrated in a short animation (***Video 1***) based on a qualitative interpretation of the LRT modeling results. Because of inherent limitations of the LRT method, this animation only illustrates the relative directional movement of the three elements without giving the correct amplitudes of motion.

The results of the LRT modeling experiment above are in good agreement with the hypothesis that the C-linker, HCND and VSD form a mechanical continuum. If mechanically connected, a similar displacement of the three domains should be elicited by applying a force on the HCND or on the S4 helix. To test this prediction, forces were applied to the HCND and to S4 while monitoring the effects on the remaining domains. As demonstrative positions to apply force on these two domains, we chose amino acids 108 on the HCND and 283 on S4 (***Figure 5A*** and ***Figure 5—figure supplement 1***). Both these residues do not engage in direct contacts between different domains, so that their perturbations should not cause a local, but rather a global effect on the mechanics of the protein. Analogous to perturbations of the C-linker (***Gross et al., 2018***; ***Figure 5—figure supplement 1***), the clustering of forces applied on the HCND or S4 was determined based on the movements of the 'shoulder' of the C-linker (C' and D' helices) (Materials and methods). The resulting movements in C-linker and HCND were then compared to the reference displacements in ***Figure 5B***. The good match observed between data in ***Figure 5B and D,E*** confirms that the same rotational movement in the protein can be elicited irrespectively on

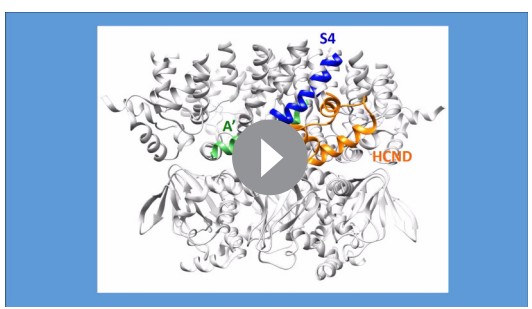

**Video 1.** Mechanical continuum between C-linker, HCND and VSD. Morphing video of HCN1 showing the concerted movement of C-linker, HCND and VSD. The structural models used for the morphing were derived from computational data obtained with the linear response theory (LRT) analysis of the HCN1 model (***Gross et al., 2018***) following application of a force vector of 400 a.u. to position A425. For simplicity, the top half of the transmembrane domain has been omitted from the illustration. The C-terminal half of S4 helix of VSD is coloured in blue; A' and B' helices of the C-linker are in green; HCND is in orange.
https://elifesciences.org/articles/49672#video1

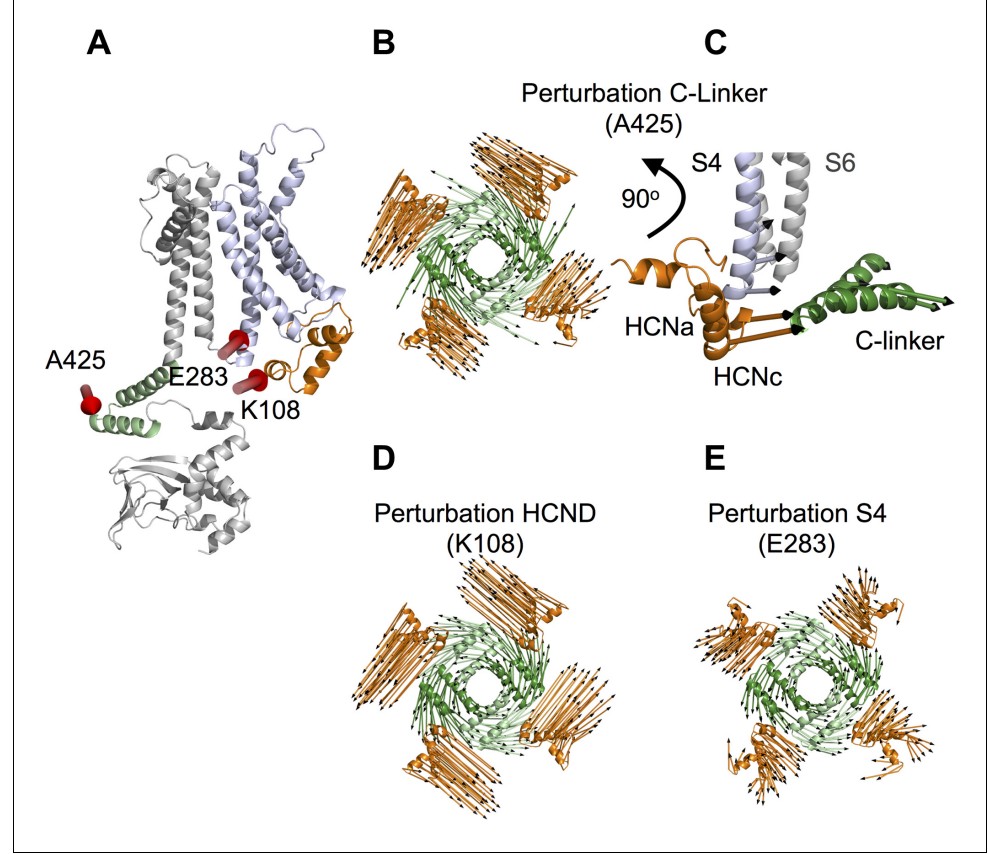

**Figure 5.** Linear response theory simulations demonstrate the mechanical coupling of C-linker, HCN domain and S4 helix in HCN1. (**A**) One HCN1 subunit (PDB_ID: 5U6O) color coded as in *Figure 1*. Positions and directions of force-displacements on residue A425 of the C-linker 'elbow', residue K108 of the HCN domain, and residue E283 of the S4 helix are indicated by red arrows. (**B**) Movements of the C-linker and HCND after perturbing the HCN1 structure at C-linker residue A425. The image shows C-linkers and HCNDs in the tetramer from an extracellular perspective. The arrows illustrate the directional movement of the domains (shown in the same color). (**C**) Side view of the structure in (B) focusing only on HCND and S4 helix from one subunit and C-linker from the opposite subunit. The selected arrows show the direction of movements of all three domains after perturbing the 'elbow'. (**D**) and (**E**) Movements of C-linker and HCND from same perspective as in (B) after perturbing the HCN1 structure at HCND residue K108 (D) or S4 helix residue E283 (E). All perturbations resulted in the same rotational movement of C-linker and HCND.

The online version of this article includes the following figure supplement(s) for figure 5:

**Figure supplement 1.** Linear response theory null model of HCN1 channel and clustering of different perturbation directions.

whether force was applied to the C-linker (*Figure 5B*), the HCND (*Figure 5D*) or the S4 helix (*Figure 5E*). This underscores that the HCND is the transmitting element in a mechanics, which connects the CNBD with the VSD, and that a disconnection of the HCND from the C-linker should abolish the allosteric effect of cAMP on the voltage dependency of HCN channels.

To test this hypothesis, we disrupted the contacts between the C-linker and the HCND in the background of HCN2 and examined gating and cAMP modulation of mutant channels. The contact of M155 (M113 in HCN1) with K464 is predicted to occur through the carbonyl oxygen, limiting our mutational strategy to the partner site on the C-linker (*Figure 6A*). In the case of the other contact, R154 (R112 in HCN1) can interact with another residue, T330, located on S4 (*Figure 6B*) in addition to E478. For these reasons, we neutralized the residues of interest in the C-linker by replacing them with an alanine (K464A and E478A).

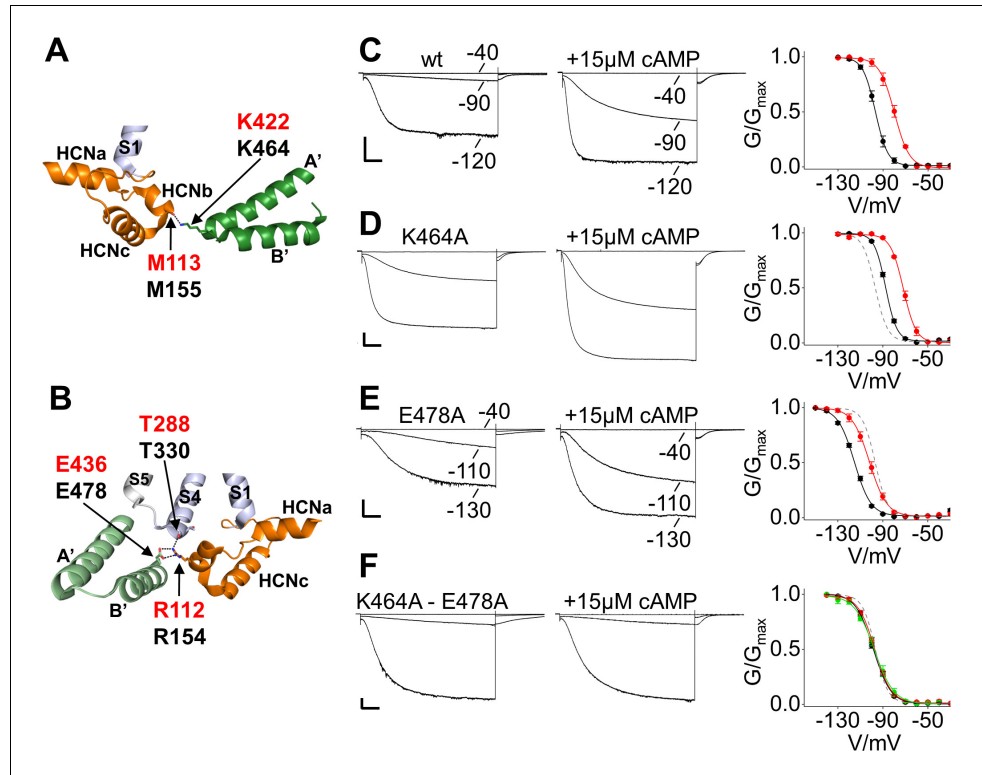

**Figure 6.** Salt bridge interactions between the HCN domain and the C-linker control cAMP effect in HCN2. (**A**) In HCN1 subunit (PDB_ID: 5U6O) (color coded as in **Figure 1**), the main chain carbonyl group of M113 in the HCN domain contacts the side chain of K422 residue on the C-linker of the opposite subunit (dark green). Salt bridges are represented as dashed black lines. Residues of interest are labelled either in red or black using hHCN1 or mHCN2 numbering, respectively. HCNa, b and c are labelled according to **Lee and MacKinnon (2017)**. (**B**) R112 of the HCN domain contacts both the side chain of E436 on the C-linker of the adjacent subunit (light green) and the main chain carbonyl group of T288 located in the S4 helix of its own subunit (violet). Representative whole-cell currents recorded from (**C**) wt, (**D**) K464A, (**E**) E478A and (**F**) K464A-E478A mutant HCN2 channels, in the absence and in the presence of cAMP. Test potentials are the same indicated in the wt traces, if not otherwise specified. Graphs on the right show mean activation curves in the absence (black circles) and in the presence of 15 µM or 100 µM of cAMP (red and green circles, respectively). Lines show data fit to a Boltzmann function (Materials and methods). Half activation potential ($V_{1/2}$) and inverse slope factor (k) values are reported in **Supplementary file 1A** together with details on statistical analysis. Values are shown as mean ± SEM. Scale bar is 200 pA x 500 ms. The fitting of the activation curve of the wt in control solution (black curve in **C**) is shown as a gray dashed line (**D-F**) for visual comparison with the activation curves of the mutant channels.

The online version of this article includes the following figure supplement(s) for figure 6:

**Figure supplement 1.** Kinetics analysis of C-linker mutants in HCN isoforms.

**Figure supplement 2.** Mutant cycle analysis of coupling between HCN4 R154A and E478A mutations.

When introduced alone, the two mutations affected channel $V_{1/2}$ in opposite ways (**Figure 6C–E**). K464A caused a shift to the right by 9.2 ± 1.3 mV, and E478A caused a shift to the left by 18.6 ± 1.4 mV (**Figure 6D,E** and **Figure 6—figure supplement 1A**). Combining the two mutations resulted in a $V_{1/2}$ value similar to the wt (100.6 ± 0.9 mV and 96.7 ± 1.2 mV for K464A-E478A and wt, respectively; not statistically different) (**Figure 6F** and **Figure 6—figure supplement 1A**). Interestingly, when we tested the effect of cAMP (15 µM) we found that the two single mutations responded to cAMP with a normal shift in the activation curve (K464A = +15.7 ± 0.9 mV; E478A = +13.5 ± 0.9 mV; wt = +16.7 ± 1.9 mV) accompanied by an increase in maximal current and an acceleration of the activation kinetics similar to that observed in the wt. However, as predicted by our hypothesis, the double mutant was no longer able to respond to cAMP (+2.3 ± 1 mV, not statistically different from control), either at the 15 µM concentration or at a higher concentration of 100 µM (**Figure 6C–F** and

*Figure 6—figure supplement 1A*). The insensitivity of the double mutant to cAMP may be due either to a dramatic decrease in ligand affinity or to a loss in efficacy that is the ability of the bound ligand to affect channel activation. The observation that the addition of cAMP is still able to affect channel deactivation in the K464A-E478A mutant, however, strongly suggests that cAMP is bound to the CNBD in the mutant and that the disruption of the HCND/C-linker interaction indeed results in a loss of the coupling between ligand binding and channel activation (*Figure 6—figure supplement 1*). Our observation that the effects measured on the deactivation pathway are independent from the effects of the same mutations on the activation pathway confirms earlier functional and modeling studies by other groups (*Altomare et al., 2001*; *Hummert et al., 2018*; *Männikkö et al., 2005*; *Wicks et al., 2011*) which indicated that opening and closing in HCN channels occur through different pathways, and that these respective pathways are differentially affected by cAMP binding.

The interaction between E478 and R154 was further investigated by mutant cycle analysis (*Chowdhury et al., 2014*). To this end we have characterized two additional HCN2 mutants, R154A and the double mutant E478A-R158A (*Figure 6—figure supplement 2A,B*) that with wt and E478A form a thermodynamic mutant cycle. With this approach, we could compare perturbation energies evaluated when E478 site was mutated in the native protein (wt) and in the background of a secondary mutation (R154). If the perturbation energies are equal, that implies the two sites do not interact or that the strength of the interaction remains the same in the closed and the open state. Unequal perturbation energies imply that the two sites interact and that the strength of the interaction is state-dependent.

*Figure 6—figure supplement 2C* shows the thermodynamic cycle in which the free energy changes associated with perturbation (ΔGp) are indicated in the scheme in kcal/mol. ΔGp were evaluated along each path from the free energy of activation of the channel during the transition closed to open (ΔGapp). Since the perturbation energies are unequal, we can conclude that the two positions are not independent (coupled) and that E478 and R154 interact. The interaction energy between the two sites, assessed by the non-additivity factor ΔΔG (see Materials and methods) is −2.0 kcal/mol.

One difference between the behaviors of the two mutants was that R154A did not show the same shift in $V_{1/2}$ that was seen with E478A.Overall, this supports our view that there is another interaction that R154 establishes besides the one with E478.

From all the above, we conclude that the C-linker interacts with the HCND at two sites, E478 and K464. Disrupting their interaction, as in the double mutant K464A- E478A, prevents cAMP effect in HCN2.

To test if the same findings hold true for the other HCN isoforms, we introduced the equivalent double mutation in HCN1 and HCN4 (*Figure 7* and *Figure 7—figure supplement 1*). We found that, in these isoforms too, disconnecting the C-linker from the HCND resulted in loss of the cAMP effects on channel activation similar to HCN2 (*Figure 7A,B*). In HCN4, the double mutant K543A-E557A showed no shift in the $V_{1/2}$ in response to 30 µM cAMP, compared to the normal ~20 mV right shift in the wt. Similar to HCN2, however, cAMP was still able to cause a significant change in the time constant of deactivation implying effective binding to the CNBD (*Figure 6—figure supplement 1*). Because the higher affinity of HCN1 for cAMP, compared to the other HCN isoforms, allows this channel to respond to endogenous levels of cAMP present in HEK293T cells, HCN1 currents measured in the whole cell configuration already have an activation curve fully shifted to the right (*Saponaro et al., 2018b*). This can be demonstrated using a mutation in the CNBD, R549E in human HCN1, which reduces the affinity for cAMP by about 1000 times (*Chen et al., 2001*). As shown in *Figure 7A*, the R549E mutation induces a leftward shift in $V_{1/2}$ of ~9 mV, which is the maximal $\Delta V_{1/2}$ response of HCN1 to cAMP binding. For this reason, we tested the response to cAMP of the double mutant K422A-E436A in HCN1 by comparing its behavior to that of the R549E mutant channel. The two channels showed a similar $V_{1/2}$ (*Figure 7A*, *Figure 7—figure supplement 1A* and *Supplementary file 1B*). This suggests that the double mutation that disconnects the C-linker from the HCND prevents the transmission of the signal exerted in HCN1 by endogenous levels of cAMP. As expected, addition of 15 µM cAMP did not affect either channels (*Figure 7A*, *Figure 7—figure supplement 1A*, and *Supplementary file 1B*).

Collectively, these data are consistent with the prediction of the LRT analysis that the HCND transmits the cAMP signal from the C-linker to the VSD and that such a mechanism is conserved in all three isoforms which show a response to cyclic nucleotides, namely HCN1, HCN2 and HCN4.

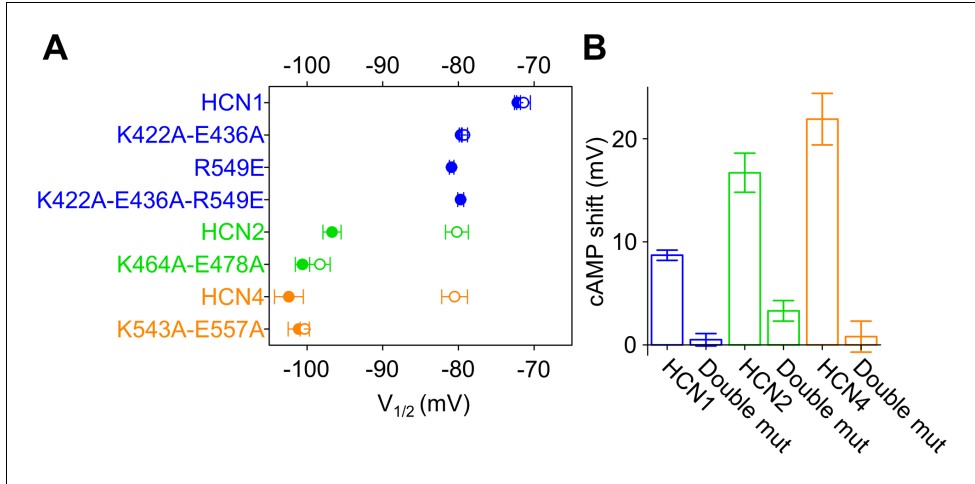

**Figure 7.** C-linker mutations preventing cAMP effect in all HCN isoforms. (**A**) Half activation voltage ($V_{1/2}$ ± SEM) values of HCN1 wt, K422A-E436A, R549E and K422A-E436A-R549E (blue symbols), HCN2 wt and K464A-E478A (green symbols), HCN4 wt and K543A-E557A (orange symbols), in control solution (filled symbols) and in the presence of 15 µM (HCN1 and HCN2) and 30 µM (HCN4) cAMP (empty symbols). (**B**) cAMP-induced shift in the $V_{1/2}$ (in mV) calculated from data in (**A**). For HCN1, the cAMP effect was calculated by subtracting wt and double mutant values from their corresponding R549E mutants. Details on statistical analysis are reported in *Supplementary file 1B*.

The online version of this article includes the following figure supplement(s) for figure 7:

**Figure supplement 1.** Properties of the C-linker double mutants in HCN1 and HCN4.

## Discussion

By investigating the amino acid contacts identified in the high-resolution structure of HCN1, we uncovered new principles of HCN channel architecture. Central to our study is the assignment of a function to the HCND, a newly discovered domain (*Lee and MacKinnon, 2017*) that is highly conserved among all HCN isoforms. Our data show that HCND ensures the correct folding and trafficking of the channel to the plasma membrane. Importantly, the data also indicate that HCND represents the structural element that allows the integration of the two different signals, voltage and cAMP, which are critical for channel gating. This is a defining property of HCN channels, which had not been previously explained in structural terms.

### Contribution of HCND to trafficking

The HCND is a folded domain found in the cytosolic N terminus of the channel, immediately preceding the first transmembrane domain, S1. The data presented here show that it is anchored to the membrane by means of two hydrophobic residues, I135 and F109 in the human HCN1 sequence (I177 and F151 in mouse HCN2), which insert into hydrophobic pockets formed by the VSD. Anchoring of the HCND to the VSD is essential for correct folding and trafficking of the channel to the plasma membrane, as demonstrated by mutant channels HCN1 I135G and HCN1 F109E which are both retained in the secretory pathway (*Figure 2—figure supplement 3*).

Our findings confirm and extend previous reports showing that HCN2 channels lacking the N terminus do not reach the plasma membrane, a phenotype attributed to the critical role of the N terminus in channel assembly (*Proenza et al., 2002*; *Tran et al., 2002*). Prior studies had further suggested that the HCNb helix contains a conserved ER export motif ([106]VNKFSL[111]) wherein L111 provides the key signal (*Pan et al., 2015*). Thus, our mutagenesis results can be interpreted in two ways: 1) disruption of the hydrophobic interactions formed by F109 and I135 may prevent correct channel assembly, resulting in a misfolded protein that is retained in the ER due to failed quality control; or 2) mutations in F109 and I135 induce conformational changes in the HCND helices which result in the [106]VNKFSL[111] ER export motif not being appropriately displayed. Either way, the observation that the HCND is critical for proper channel expression at the plasma membrane represents a

challenge towards any functional study of the HCN domain, and calls for the need to delete/mutate the HCND after the channel has reached the plasma membrane.

## Contribution of HCND to gating

Our study uncovered a major contribution of the HCND to channel function, by providing evidence that the HCND is the structural element that functionally couples the VSD with the C-linker/CNBD. By forming a mechanical continuum between the voltage sensing and ligand binding elements, the HCND is able to mutually transmit conformational information directly between the CNBD and the VSD, bypassing the pore.

Coupling of the VSD to the CNBD via the HCND is achieved through distinct intra-molecular interactions. Contact between the HCND and the voltage sensor is mediated by hydrophobic interactions, wherein an aromatic side chain (F109/F151 in HCN1 and HCN2 respectively) inserts into a hydrophobic pocket formed by S1 and S4 residues. In contrast, contact between the HCND and the C-linker is established by means of salt bridges in a hydrophilic environment. The HCND/C-linker contacts show an interesting bifurcated interaction pattern. Two adjacent amino acids on the HCND form a common point of interaction from which they branch to form salt bridges with the A′-helix and B′-helix of C-linkers from the adjacent and opposite subunits, respectively. This structural arrangement, which joins the C-linkers of two different subunits to a common attachment point in the HCND of a third subunit, has important consequences for the transmission of the movement within the protein. Previous experimental and computational studies have indicated that binding of cAMP generates an iris-like movement in the 'gating ring' portion of the C-linker, which is formed by the A′ and B′ helices (*Craven and Zagotta, 2004*; *Gross et al., 2018*; *Lee and MacKinnon, 2017*; *Marchesi et al., 2018*; *Shin et al., 2004*; *Weißgraeber et al., 2017*). This movement occurs in the plane parallel to the membrane. Due to the nature of the attachment between the C-linker helices and HCND, this planar rotational movement is translated into an upward rotational movement of the HCND. This, in turn, results into a tilting of the S4 domain (*Video 1*) due to the tight hydrophobic interactions formed between the HCND and the VSD.

These new structural and computational insights provide a coherent model for the allosteric regulation of HCN channels. Our experimental data also show that such mechanical link is conserved between the different HCN isoforms, thus providing a general mechanism for the modulation by voltage and cAMP, a feature common to the HCN1, HCN2 and HCN4 isoforms.

## Control of voltage dependence

Multiple lines of evidence indicate that HCND contributes to the control of the HCN channels' voltage dependence. For one, destabilizing the interaction between the HCND and the loop connecting S2 with S3 of the VSD (mutants HCN2 I176A and HCN2 L250A) led to a distinct decrease in the time-dependent component of the HCN current, accompanied by a corresponding increase in the instantaneous component. This indicates a loss in the voltage dependence of opening and closing, resulting in a 'leaky' channel. The most revealing information, however, comes from our analysis of the F151 residue, aimed at disrupting the hydrophobic interaction between the HCND and S4 of the VSD. The strong shift in the voltage dependence of the F151E mutation towards more depolarized potentials (+ 34 mV) implies that the HCND normally keeps the VSD in a position, which is unfavorable for channel opening. In other words, the HCND acting via F151 exerts an inhibitory action on the VSD, reminiscent of the previously highlighted inhibitory effect of CNBD on the channel (*Wainger et al., 2001*).

Such a model is also supported by the behavior of less severe mutants (e.g. F151V/A), in which the hydrophobic contact is maintained but weakened. In these mutants, the $V_{1/2}$ values are progressively right shifted. This is in good agreement with a model in which a reduced impact of HCND on the VSD lowers the energy barrier for channel opening.

## Integration of voltage dependence and allosteric modulation by cAMP

The impact of the HCND on the VSD function, and the central position occupied by the HCND between the C-linker/CNBD and VSD, also provide a coherent explanation for the mechanism of allosteric regulation of HCN channels by cAMP. Both computational and experimental data suggest that the conformational change in the C-linker, which is generated by the binding of cAMP, is

transmitted via the HCND to the VSD, lowering the energy barrier for channel opening. As long as the HCND is connected to the hydrophobic pocket formed by S4, cAMP is still able to shift the voltage dependence of the channel to more positive voltages as in mutants HCN2 F151V and F151A. In these circumstances, conformational information, which arrives from the CNBD, is translated into a reorientation of the VSD under the influence of the HCND. In contrast, cAMP has no longer an impact on the voltage dependence of the channel once the contact between the VSD and the HCND is completely disrupted as in mutant HCN2 F151E.

Just like the contact between VSD and the HCND is important for the allosteric regulation of gating, the contact between the HCND and the C-linker is equally essential. In line with the idea of a mechanic continuum between the three elements, we find that mutants, which disrupt the contacts between the C-linker and the HCND, modulate the gating behavior of the channels. Disruption of these contacts eliminated the effect of cAMP on channel activation (e.g. HCN2 K464A-E478A or HCN4 K543-E557). This is in good agreement with the model in which the HCND serves as a coupling and transmission element between CNBD and VSD.

The mutational analysis conducted in parallel in HCN1 and HCN2 (and HCN4) gave more consistent results in the case of C-linker/HCND interaction than of HCND/VSD highlighting that isoform-specific responses to cAMP emerge from the mechanisms that connect the cytosolic to the TM domains.

Our results confirm a large body of published experimental findings, and at the same time open a new and unexpected perspective on the gating mechanism of HCN channels. It is well established that cAMP alone does not open the pore in the absence of voltage. This experimental finding has been attributed to the fact that the voltage sensor exerts an inhibition on the ability of the cAMP signal to open the pore. Our data now show that removal of this inhibition requires both voltage and the physical connection of the VSD to the C-linker mediated by the HCND. In this scenario, the rotational movement of the C-linker following binding of cAMP to the CNBD translates via the HCND into a small movement of the VSD that may preactivate the channel by lowering the voltage required to move the sensor (*Kusch et al., 2010*). Thus, we suggest that the autoinhibitory effect of the CNBD, which constrains gating to more hyperpolarized potentials (*Wainger et al., 2001*), is directly mediated by the HCND. The latter might then prevent or promote the extent of the movement of the VSD depending on the bound or unbound state of the CNBD.

The model we propose is in good agreement with functional and computational data published in the literature. Thon and collaborators proposed that ligand binding to the CNBD promotes movement of the voltage sensor in a concentration dependent manner, including when the channel is not preactivated by voltage (*Kusch et al., 2010*; *Thon et al., 2015*). Moreover, channel preactivation by voltage causes an increase in cAMP binding affinity within the CNBD prior to pore opening, further demonstrating that VSD and CNBD are functionally coupled in such a manner as to bypass the pore (*Kusch et al., 2010*). Our findings are also consistent with the gating model suggested by *Bell et al. (2004)*: these authors proposed that tilting movements in S4 and surrounding transmembrane segments in response to hyperpolarizing voltages cause the formation of a water-filled crevice (gating canal) at the C-terminal end of the S4 helix. The motion of the latter was proposed to be coupled to opening of the activation gate of the channel. Intriguingly, the hydrophobic pocket formed by the S4 helix, which connects to F151 in the HCND, is located precisely at the level of the C-terminal tail of S4, where the gating canal is expected to form. In light of existing data, we can therefore speculate that any influence of the HCND on the conformation of this hydrophobic pocket might lower or increase the energy barrier for the formation of the internal gating canal. In this way, the movement of the HCND may facilitate or antagonize the channel opening.

Finally, our finding that the cAMP-induced movement of the C-linker does not facilitate pore opening if the C-linker is disconnected from the HCND, confirms the postulated inhibitory action of the VSD on the C-linker and explains why cAMP doesn't open the pore without voltage. Given the relatively high sensitivity of HCN channels to cAMP (with $K_d$ in the submicromolar range), this mechanism would protect the cell from occasional channel opening in response to physiological or pathological oscillations of cAMP concentrations.

The mechanical continuum formed by the VSD, HCND and C-linker/CNBD through the intramolecular interactions established between these three elements thus provides a physiologically relevant model for the allosteric modulation of HCN channels by voltage and cAMP. The newly discovered role of HCND provides a rationale for mutations identified in the HCND in patients with early

infantile epileptic encephalopathy, a severe form of childhood epilepsy (*Marini et al., 2018*; *Nava et al., 2014*).

In conclusion, we show here that the effect of cAMP on voltage dependent channel activation is mediated in HCN channels by the HCND. The HCND is sandwiched between the VSD and the C-linker and establishes through intra-molecular interactions a mechanical continuum between the three elements. This molecular ensemble provides a coherent model for the allosteric modulation of HCN channels by voltage and cAMP.

# Materials and methods

## Key resources table

| Reagent type (species) or resource | Designation | Source or reference | Identifiers | Additional information |
|---|---|---|---|---|
| Gene (human) | HCN1 | Xention Ltd. (Cambridge,UK) | NM_021072.3 | |
| Gene (mouse) | HCN2 | PMID: 11331358 | NM_008226.2 | |
| Gene (rabbit) | HCN4 | PMID: 10212270 | NM_001082707 | |
| Strain, strain background (*E. coli*) | Stbl2 | Thermo Fisher Scientific | | |
| Cell line (human) | HEK 293T | ATCC (Authenticated by STR profiling) | RRID:CVCL_0063 | Tested for mycoplasma: negative result |
| Recombinant DNA reagent | pcDNA 3.1 (plasmid) | Clontech Laboratories | | |
| Recombinant DNA reagent | pCI (plasmid) | Promega | | |
| Recombinant DNA reagent | pEGFP (plasmid) | Clontech Laboratories | | |
| Transfected construct (human) | HCN1 (cDNA) | Xention Ltd (Cambridge,UK) | | All HCN1 mutants transfected in the paper were obtained starting from this wt cDNA. For further details please see Materials and methods |
| Transfected construct (mouse) | HCN2 (cDNA) | Laboratory of Steven A. Siegelbaum | | All HCN2 mutants transfected in the paper were obtained starting from this wt cDNA. For further details please see Materials and methods |
| Transfected construct (rabbit) | HCN4 (cDNA) | PMID: 10212270 | | All HCN4 mutants transfected in the paper were obtained starting from this wt cDNA. For further details please see Materials and methods |
| Transfected construct (human) | GFP-HCN1 (cDNA) | This paper | | All GFP-HCN1 mutants transfected in the paper were obtained starting from this wt cDNA. For further details please see Materials and methods |

*Continued on next page*

*Continued*

| Reagent type (species) or resource | Designation | Source or reference | Identifiers | Additional information |
|---|---|---|---|---|
| Transfected construct (mouse) | GFP-HCN2 (cDNA) | PMID: 15564593 | | All GFP-HCN2 mutants transfected in the paper were obtained starting from this wt cDNA. For further details please see Materials and methods |
| Commercial assay or kit | QuickChange Lightning Site-Directed Mutagenesis Kit | Agilent | | |
| Commercial assay or kit | Exprep Plasmid SV kit | GeneAll | | |
| Commercial assay or kit | Thermo Scientific TurboFect Transfection Reagent | Thermo Fisher Scientific | | |
| Chemical compound, drug | Adenosine 3', 5'-cyclic monophosphate (cAMP) | SIGMA | | |
| Chemical compound, drug | CellMask Deep Red Plasma membrane Stain | Thermo Fisher Scientific | | |
| Software, algorithm | pClamp - Clampfit | Molecular Devices | RRID:SCR_011323 | Version 10.7 |
| Software, algorithm | pClamp - Clampex | Molecular Devices | RRID:SCR_011323 | Version 10.7 |
| Software, algorithm | EZ-Patch | Elements srl https://elements-ic.com/ | | Version 1.0.0 |
| Software, algorithm | Gromacs | http://www.gromacs.org/ | RRID:SCR_008395 | Version 9.22 |
| Software, algorithm | Modeller | http://salilab.org/modeller/modeller.html | RRID:SCR_014565 | Version 2018.x |
| Software, algorithm | Python | http://www.python.org | RRID:SCR_008394 | Version 3.7.3 |
| Software, algorithm | Pymol | https://pymol.org | RRID:SCR_000305 | Version 2.2.0 |
| Software, algorithm | Fiji ImageJ | http://fiji.sc/ | | |

## Constructs

HCN genes were cloned in pcDNA3.1 (hHCN1) or in pCI (rbHCN4 and mHCN2) expression vectors. For fluorescent microscopy experiments, HCN1 channels were cloned in pcDNA 3.1 with an enhanced green fluorescent protein (EGFP) tag fused in frame to their N-termini. HCN2 channels were cloned in pEGFP-C1 and additionally contained an extracellular HA-tag sequence. The HA-tag sequence is inserted between amino acids G284 and I285 by means of a 7aa long linker (linker-HA-tag: ISAYGIT-YPYDVPDYA). Site-directed point mutations were introduced in the hHCN1, mHCN2 and rbHCN4 genes using the QuickChange Lightning (Agilent Technologies) kit following the specifications recommended by the manufacturer. All constructs were verified by full-length sequencing. Stbl2 competent cells (Invitrogen) were used to amplify the plasmid DNA, which was then extracted using Exprep Plasmid SV kit (GeneAll) according to the manufacturers recommended protocol.

## Cell culture and transfection

HEK293T cells were cultured in Dulbecco's modified Eagle's medium (Euroclone) supplemented with 10% fetal bovine serum (Euroclone), 1% Pen Strep (100 U/mL of penicillin and 100 µg/ml of strepto-mycin) and stored in a 37°C humidified incubator with 5% $CO_2$. Every two or three days cells were trypsinized and split to avoid overgrowth. After 20–25 splits cells were discarded and a new fresh

line was thawed to substitute the old one. Cells were grown in a 25 mm$^2$ flask and transferred in 35 mm Petri dishes (Sarstedt) the day before transfection. When ~ 70% confluent, HEK293T cells were transiently transfected with wild-type and/or mutant cDNA using Turbofect transfection reagent (Thermo Fisher) according to the manufacturers recommended protocol.

## Electrophysiology in HEK cells and data analysis

For each 35 mm Petri dish 1 µg or 0.5 µg of the HCN-containing vector and 0.3 µg of EGFP-containing plasmid (pmaxGFP, AmaxaBiosystems) were used. 30–72 hr after the transfection the cells were dispersed by trypsin treatment. Green fluorescent cells were selected for patch-clamp experiments at room temperature (about 25℃). Currents were recorded in whole-cell configuration either with an Axopatch 200B amplifier (Molecular Devices, CA, USA) or with a ePatch amplifier (Elements, Cesena, Italy); data acquired with the Axopatch 200B amplifier were digitized with an Axon Digidata 1550B (Molecular Devices, CA, USA) converter. All data were analysed off-line with Axon pClamp 10.7. Patch pipettes were fabricated from 1.5 mm O.D. and 0.86 I.D. borosilicate glass capillaries (Sutter, Novato, CA, USA) with a P-97 Flaming/Brown Micropipette Puller (Sutter, Novato, CA, USA) and had resistances of 3–6 MΩ. The pipettes were filled with a solution containing: 10 mM NaCl, 130 mM KCl, 1 mM egtazic acid (EGTA), 0.5 mM MgCl$_2$, 2 mM ATP (Magnesium salt) and 5 mM HEPES–KOH buffer (pH 7.4). The extracellular bath solution contained 110 mM NaCl, 30 mM KCl, 1.8 mM CaCl$_2$, 0.5 mM MgCl$_2$ and 5 mM HEPES–KOH buffer (pH 7.4). Where indicated, different volumes of Adenosine 3′,5′-cyclic monophosphate (cAMP, Sigma-Aldrich) were added to the pipette solution from a previously prepared stock solution in order to obtain different concentrations. The stock solution was prepared solving the powder in milliQ water in order to obtain a final concentration of 100 mM and adjusting the pH to 7. Single-use aliquots were made and stored at −20℃ until the day of the experiment.

For channel activation, hyperpolarizing steps of variable duration, sufficient to reach steady-state activation at all voltages, were applied from a holding potential of −30 mV and current tails were measured upon return to a fixed voltage (−40 mV for all the isoforms). The duration and the number of the steps used to activate the channels were adjusted for the different HCN isoform. Whole-cell measurements of HCN channels were performed using the following voltage-clamp protocol depending on the HCN isoform measured: for HCN1, holding potential was −30 mV (1 s), with steps from −20 mV to −120 mV (10 mV interval, 3.5 s) and tail currents recorded at −40 mV (3 s); for HCN2, holding potential was −30 mV (1 s), with steps from −40 mV to −130 mV (10 mV interval, 5 s) and tail currents recorded at −40 mV (5 s); for HCN4, holding potential was −30 mV (1 s), with steps from −30 mV to −165 mV (15 mV interval, 5 s) and tail currents were recorded at −40 mV (5 s). Only cells in which a 1 GΩ seal or higher was achieved were kept for analysis. Patch-clamp currents were acquired with a sampling rate of 5 kHz and lowpass filtered at 1 kHz.

Mean activation curves were obtained by fitting maximal tail current amplitude, plotted against the voltage step applied, with the Boltzmann equation

$$y = 1/[1 + \exp((V - V_{1/2})/k)] \tag{1}$$

where V is voltage, y the fractional activation, $V_{1/2}$ the half-activation voltage, and k the inverse-slope factor = -RT/zF, using Originpro software (Originlab, Northampton, MA, USA). Mean $V_{1/2}$ values were obtained by fitting individual curves from each cell to the Boltzmann equation and then averaging all the obtained values.

Activation and deactivation time constants (τ) were obtained by fitting a single exponential function,

$$I = I_0 \exp(-t/\tau) \tag{2}$$

to current traces obtained with the activation protocol described above. Deactivation time constants were obtained by fitting tail currents collected at −40 mV after a fully activation pulse at −130 mV (HCN2), −135 mV (HCN4) and −120 mV (HCN1).

All measurements were performed at room temperature. Mean $V_{1/2}$ values derived by the Boltzmann fitting were compared using one-way ANOVA followed by Fisher's test or using Student's t-test. Significance level was set to p = 0.05. All data are presented as mean ± standard error of the mean (SEM).

## Confocal microscopy

Cell fluorescence was measured using a Nikon Eclipse-Ti inverted confocal microscope interfaced with an A1 series of confocal laser point scanning system for excitation at 405, 488, 561 and 640 nm. Glass bottom Petri dishes containing HEK293T cells were transiently transfected with EGFP-mHCN2-HA or EGFP-hHCN1. Fluorescence analysis was carried out 24 hr after transfection on living cells. The plasma membrane was stained with CellMaskTM Deep Red from Invitrogen according the manufacturers protocol. The samples were observed with a $60 \times 1.4$ NA oil immersion objective (Nikon System). The pinhole aperture was set to 1.0 Airy. The images were collected using low excitation power (488 and 640 nm) at the sample and acquiring the emission range through bandpass filters 525/50 and 700/75 for EGFP and CellMaskTM emission, respectively, by means of built-in GaAsP PMT detectors of the confocal microscope.

## Thermodynamic mutant cycle analysis

The parameters k and $V_{1/2}$ were determined from fits to current-voltage relationships as described above. The amount of free energy required to shift the inward rectifying channel from the closed to the open state was calculated as $\Delta Gapp = (RT/k) V_{1/2}$, where R and T have the usual meaning and $V_{1/2}$ is the half-activation voltage and k is the inverse slope factor = -RT/zF. The perturbation in free energy of the mutant channel relative to the WT was calculated as $\Delta Gp = \Delta(RT/k)V_{1/2}$. Coupling of nonadditive free energy was calculated as $\Delta\Delta G = \Delta Gapp\ wt + \Delta Gapp\ R154A\text{-}E478A - \Delta Gapp\ R154A - \Delta Gapp\ E478A$ (*Chowdhury et al., 2014*). $\Delta Gapp$, $\Delta Gp$ and $\Delta\Delta G$ are in kcal/mol.

## Molecular dynamics simulation

MD simulations were carried out using the GROMACS 2018 software suite in combination with the CHARMM36m forcefield in its July2017 revision (*Abraham et al., 2015*; *Huang and MacKerell, 2013*; *Pronk et al., 2013*; *Van Der Spoel et al., 2005*). The Cryo-EM structure of HCN1 without complexed cAMP (PDB: 5U6O) embedded into a preequilibrated palmitoyloleoylphosphatidyl-cholin (POPC) membrane was used (*Lee and MacKinnon, 2017*) In-silico point mutations were introduced using Modeller 9.19 (*Sali and Blundell, 1993*). Symmetry restraints were applied on all residues to maintain the four-fold symmetry of the protein. Titratable residues were then protonated according to their estimated pKa value using PROPKA3 (*Olsson et al., 2011*). Details of the simulation setup have been described previously (*Marini et al., 2018*). The total number of molecules in the simulation systems are listed in *Supplementary file 1C*. Analyses were performed using GROMACS tools and biotite 0.11.1 (*Kunzmann and Hamacher, 2018*).

## Linear response theory (LRT)

The LRT calculations were carried out as described previously (*Gross et al., 2018*; *Weißgraeber et al., 2017*) using a curated structure of the cAMP-free HCN1 channel (PDB ID: 5U6O) (*Lee and MacKinnon, 2017*). This structure was reduced to a heterogeneously parametrized anisotropic network model (ANM) (*Atilgan et al., 2001*) with a cutoff for connected residues of 13 Å. The ANM was then perturbed simultaneously on all four monomers at defined residues (either A425, or K108, or E283) by using 1000 external force vectors with a force strength of 1600 in arbitrary units (a.u). Afterwards, clustering of the random force directions into four clusters was done based on the displacements of residues 446–465 (C' and D' helices), which represent the 'shoulder' of the C-linker. The decision to use four clusters was based on the comparison of log values of maximal within-cluster sum of squares (maximum withinss) from k-means clustering as a function of number of clusters (*Hartigan and Wong, 1979*) which was explained in detail in *Gross et al. (2018)*. The four clusters obtained in each experiment are visualized in *Figure 5—figure supplement 1* as cluster of red, blue, yellow or green vectors. For each cluster, one representative force direction was manually selected from the cluster center and the resulting displacements in the protein were then analyzed. We previously found that forces from the yellow cluster of vectors applied to the 'elbow' of the C-linker (residue A425) cause conformational changes in the protein, which are similar to those elicited by cAMP binding (*Gross et al., 2018*). Therefore, we used the corresponding rotational movements of the C-linker and HCND as a reference and searched for the direction of force vectors applied to the HCND (residue K108) and S4 segment (residue E283), respectively, which would cause

similar displacements in the protein. All such clusters are labeled in yellow in the illustrations presented in *Figure 5—figure supplement 1*.

Figures shown in the text were prepared using the PyMOL Molecular Graphic System (http://www.pymol.org/).

## Acknowledgements

Calculations required for MD simulations were conducted on the Lichtenberg high performance computer of the Technische Universität Darmstadt. The authors would like to thank the Hessian Competence Center for High Performance Computing, funded by the Hessen State Ministry of Higher Education Research and the Arts, for helpful advice. This research was funded by FONDA-ZIONE CARIPLO, *2014–0796* and H2020 - ERC-2015-AdG 695078-noMAGIC.

## Additional information

### Funding

| Funder | Grant reference number | Author |
|---|---|---|
| Fondazione Cariplo | 2014-0796 | Anna Moroni |
| H2020 European Research Council | ERC-2015-AdG 695078-noMAGIC | Anna Moroni |

The funders had no role in study design, data collection and interpretation, or the decision to submit the work for publication.

### Author contributions

Alessandro Porro, Conceptualization, Data curation, Formal analysis, Investigation, Methodology; Andrea Saponaro, Conceptualization, Investigation, Methodology; Federica Gasparri, Matteo Pisoni, Gerardo Abbandonato, Data curation, Formal analysis; Daniel Bauer, Christine Gross, Data curation, Formal analysis, Investigation; Kay Hamacher, Conceptualization, Supervision, Methodology; Bina Santoro, Conceptualization; Gerhard Thiel, Conceptualization, Supervision; Anna Moroni, Conceptualization, Supervision, Funding acquisition

### Author ORCIDs

Alessandro Porro ![iD] https://orcid.org/0000-0003-4845-6165
Andrea Saponaro ![iD] http://orcid.org/0000-0001-5035-5174
Gerardo Abbandonato ![iD] http://orcid.org/0000-0001-7247-051X
Bina Santoro ![iD] http://orcid.org/0000-0002-4277-1992
Anna Moroni ![iD] https://orcid.org/0000-0002-1860-406X

### Decision letter and Author response

Decision letter https://doi.org/10.7554/eLife.49672.sa1
Author response https://doi.org/10.7554/eLife.49672.sa2

## Additional files

### Supplementary files

- Supplementary file 1. (**A**) Fitting parameters of the activation curves in HCN2 (*Figures 2,4,6,7*). From left to right: half-activation voltage ($V_{1/2}$), inverse slope factor (k) obtained by fitting data to a Boltzmann function (Material and methods) in absence or presence of cAMP; n = number of cell tested in each condition; cAMP-induced shift in $V_{1/2}$; number of cells that expressed a measurable HCN current. *$p<0.05$ by One-way ANOVA with Fisher's test compared to wt HCN2; [§]$p<0.05$ by Student's T-test compared to control condition (without cAMP); n.s. not statistically different; n.t. not tested; n.d. not detectable. cAMP concentration was 15 µM in all cases, except for last row (HCN2 K464-E478A[#], 100 µM cAMP). (**B**) Fitting parameters of the activation

curves in HCN1 and HCN4 (*Figures 2,7*). From left to right: half-activation voltage ($V_{1/2}$), inverse slope factor (k) obtained by fitting data to a Boltzmann function (Material and methods) in absence or presence of cAMP; n = number of cell tested in each condition; cAMP-induced shift in $V_{1/2}$; number of cells that expressed a measurable HCN current. *$p<0.05$ by One-way ANOVA with Fisher's test compared to wild-type HCN1 or HCN4; §$p<0.05$ by Student's T-test compared to control condition (without cAMP); n.s. not statistically different; n.t. not tested; n.d. not detectable. cAMP concentration used for HCN1 and HCN4 was 15 µM and 30 µM respectively. (**C**) Number of molecules for each simulation of molecular dynamics performed on HCN1. POPC: 1-palmitoyl-2-oleoyl-glycero-3-phosphocholine; TIP3P: water model; K: $K^+$ ion; CL: $Cl^-$ ion

• Transparent reporting form

## Data availability

All data analyzed during this study are included in the manuscript and supporting files. Source data files for LRT analysis and MD simulations have been deposited in Dryad and are available at https://doi.org/10.5061/dryad.rn85375.

The following dataset was generated:

| Author(s) | Year | Dataset title | Dataset URL | Database and Identifier |
|---|---|---|---|---|
| Anna Moroni | 2019 | Date from: The HCN domain couples voltage gating and cAMP response in Hyperpolarization-activated Cyclic Nucleotide-gated channels | https://dx.doi.org/10.5061/dryad.rn85375 | Dryad Digital Repository, 10.5061/dryad.rn85375 |

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
