## [Decision Letter]

**Acceptance summary:**

In this paper Porro and colleagues address the molecular mechanism by which binding of cAMP modulates the voltage sensitivity of hyperpolarization-activated cyclic nucleotide-gated (HCN) channels. They examine the role of the N-terminal HCN domain (HCND), which was identified in recent cryo-EM structures as a structural segment wedged in between the channel's voltage sensor domain (VSD) and its C-terminal cytosolic domains (C-linker ring and cyclic nucleotide-binding domain (CNBD)). The authors find that perturbing hydrophobic interactions between the HCND and the VSD impairs correct folding and trafficking. Moreover, disrupting interactions either between HCND and VSD or between HCND and C-linker ring abolishes the stimulatory effect of cAMP on channel activation. These observations identically apply to three different HCN isoforms, human HCN1, mouse HCN2, and rabbit HCN4. The authors conclude that the HCND serves to transmit the conformational change induced by cAMP binding to the transmembrane VSD. These findings provide a mechanistic explanation for the link between voltage- and cAMP-induced activation, which is an important step forward in understanding structure-function relationships of HCN channels.

**Decision letter after peer review:**

Thank you for submitting your article "The HCN domain couples voltage gating and cAMP response in Hyperpolarization-activated cyclic nucleotide-gated channels" for consideration by *eLife*. Your article has been reviewed by three peer reviewers, including László Csanády as the Reviewing Editor and Reviewer #1, and the evaluation has been overseen by Olga Boudker as the Senior Editor. The following individuals involved in review of your submission have agreed to reveal their identity: John Bankston (Reviewer #2); Marcel P. Goldschen-Ohm (Reviewer #3).

The reviewers have discussed the reviews with one another and the Reviewing Editor has drafted this decision to help you prepare a revised submission.

All three reviewers have found the work interesting and the data of high quality, but some concerns were raised which will need to be addressed before publication. Specifically, interpretation of single-mutant data in terms of disruption of one specific interaction needs further experimental support (see sections 1 and 2) below).

Essential revisions:

1) In the HCN2 background, the authors pinpoint interacting residue pairs I176-L250 and F151-Y180, R154-E478, and M155-K464. For each of the above four pairs the authors mutate only one of the two involved side chains, and interpret the resulting phenotypes to reflect the consequence of loss of interaction between the target pair. While this interpretation might be correct, interpreting the effect of a single mutation in terms of loss of a specific interaction is questionable. The conclusions would be strengthened by performing conventional mutant cycles, i.e., by mutating both interacting residues individually and in combination. Non-additivity of mutation-induced functional effects in the double mutant would support (and energetically quantify) the functional relevance of the proposed interaction.

E.g., in addition to the already characterized WT and E478A constructs, the authors should also characterize R154A and R154A/E478A. These four constructs form a thermodynamic mutant cycle. The free energy difference between the open and the closed state at zero mV (deltaGo-c) can be expressed as deltaGo-c = -z*F*V1/2 (or deltaGo-c = -(R*T/k)*V1/2, using the authors' terminology). Energetic effects of the two single mutations and of the double mutation should be quantitated as deltadeltaGo-c = -(R*T/k)*deltaV1/2. Non-additive effects of the two mutations in the double mutant would support the authors' hypothesis and at the same time quantify the energetic contribution of the R154-E478 salt bridge to the stabilization of the open state. Alternatively, a charge-swap between positions 154 and 478 (R154E, E478R, R154E/E478R) could be attempted, but this would require characterizing three novel constructs. We leave it to the authors' discretion to try one of those approaches.

2) F151E was the only mutation shown to alter cAMP regulation of V1/2 at the HCND-VSD interface. To support the idea that this results from a charged substitution perturbing a hydrophobic pocket, the authors should test whether other charged substitutions (D,K,R) have a similar effect.

3) Most of the mutations in the first interaction site (Figure 2) result in channels that don't make it to the membrane, however, the I176A and L250A mutants both do something interesting to gating. Is the voltage dependence of these channels lost or just shifted? What do the IVs look like? Did you try to hold more depolarized to close the channels?

4) In the second hydrophobic interaction site (Figure 3, Figure 4), you use MD to look at how the local structure changes when the central PHE is mutated and show a disruption of the structure. This is reported as a change in distance between the mutated residue and a nearby TYR. Some clarification about the relationship between the simulation and the data are needed. The three largest changes in the simulation are caused by the mutations to VAL, ALA, and GLU which all perturb the measured distance by the same amount. However, the functional data reports a dramatically different voltage dependence between these three mutants and in the last case no sensitivity to cAMP. It is unclear how the MD simulation relates and enhances these data.

5) In a number of places in the hydrophobic site (I134A versus I176A or F109 versus F151) there is a substantial difference between your results in HCN1 versus HCN2, whereas the C-linker HCND interaction results are quite consistent. Some comment on the ubiquity of this mechanism in the HCN family might be warranted.

6) A little more discussion of linear response theory and what the modeling can tell you and its limitations might be helpful. We are unfamiliar with this approach and a broad audience might be as well. If you break the putative bonds/salt bridges does the model tell you that the force is not transmitted as well?

---

## [Author Response]

Essential revisions:1) In the HCN2 background, the authors pinpoint interacting residue pairs I176-L250 and F151-Y180, R154-E478, and M155-K464. For each of the above four pairs the authors mutate only one of the two involved side chains, and interpret the resulting phenotypes to reflect the consequence of loss of interaction between the target pair. While this interpretation might be correct, interpreting the effect of a single mutation in terms of loss of a specific interaction is questionable. The conclusions would be strengthened by performing conventional mutant cycles, i.e., by mutating both interacting residues individually and in combination. Non-additivity of mutation-induced functional effects in the double mutant would support (and energetically quantify) the functional relevance of the proposed interaction.E.g., in addition to the already characterized WT and E478A constructs, the authors should also characterize R154A and R154A/E478A. These four constructs form a thermodynamic mutant cycle. The free energy difference between the open and the closed state at zero mV (deltaGo-c) can be expressed as deltaGo-c = -z*F*V1/2 (or deltaGo-c = -(R*T/k)*V1/2, using the authors' terminology). Energetic effects of the two single mutations and of the double mutation should be quantitated as deltadeltaGo-c = -(R*T/k)*deltaV1/2. Non-additive effects of the two mutations in the double mutant would support the authors' hypothesis and at the same time quantify the energetic contribution of the R154-E478 salt bridge to the stabilization of the open state. Alternatively, a charge-swap between positions 154 and 478 (R154E, E478R, R154E/E478R) could be attempted, but this would require characterizing three novel constructs. We leave it to the authors' discretion to try one of those approaches.

We thank the reviewers for their valuable suggestion. We have chosen to perform double mutant cycle analysis on the R154-E478 couple. The results, newly added as Figure 6—figure supplement 2, confirm that the two positions are not independent. From the perturbation energies we calculated a ΔΔG value of -2 kcal/mol, indicating coupling between the two mutations and supporting the proposed model.

2) F151E was the only mutation shown to alter cAMP regulation of V1/2 at the HCND-VSD interface. To support the idea that this results from a charged substitution perturbing a hydrophobic pocket, the authors should test whether other charged substitutions (D,K,R) have a similar effect.

We tested the three other substitutions F151D, F151K and F151R both by ephys and MD simulations and the results are summarized in the Figure 4—figure supplement 2 and in the text in subsection “Second hydrophobic pocket”. Here is a short summary of their behaviour:

F151D

Ephys experiments show that substitution F151D resulted in a channel that gives no measurable currents (0/30 GFP+ cells). We presume that this happens because the mutant channel does not reach the membrane, a behaviour already observed for F151E for which we could measure a current in about 10% only of the GFP+ cells (3/30 GFP+ cells). MD simulation of F151D mutant shows that the aspartate side chain does not stay inside the hydrophobic pocket and that the residue is fully hydrated.

F151K

This mutant shows a similar behaviour to the previous one. It is very difficult to find a cell that expresses a measurable current; we could record a small current in 3 cells over 30 GFP+ cells but the currents were not large enough to allow an analysis of V_1/2_ and to test a cAMP response. MD simulations shows that the side chain of this residue is not inside the hydrophobic pocket, supporting our hypothesis of altered folding and trafficking.

F151R

When we introduce this mutation, we were at first surprised to find no difference between mutant and wt channel with respect to the measured parameters, V_1/2_ and cAMP-induced shift. Thinking about this conundrum we realized that arginine has distinct features that become relevant when inserted into lipid environment. This has also been highlighted to explain, for instance, the large preference for Arg over Lys as a gating charge carrier in K_v_ channels. The peculiarity of Arg relates to the hydration properties of the two side chain ions, guanidinium for Arg and aliphatic amino for Lys, that makes dehydration energetically less costly in the case of Arg (Mason PE et al., 2003). Moreover, because of the long length of its side chain (3 CH2 groups), Arg can also cover conformational space more extensively than Lys, (and Glu or Asp) increasing the probability of placing the guanidinium moiety near polar protein atoms (Harms et al., 2011, PNAS, 108: 18554; Armstrong et al., 2016). MD simulations of this mutant confirmed that the arginine side chain inserts deep inside the hydrophobic pocket to form an hydrophilic interaction with D183, a residue on S1 that lines the pocket from the back.

The reduced frequency with which we find the channel at the membrane in patch experiments (F151R 10/24, wt 28/28 cells) suggests nonetheless that this mutant indeed has also some problems with respect to its trafficking, compared to the wt. This is reflected in the low but still measurable solvation of this residue calculated from by MD simulation indicating that in a fraction of channels the side chain is exposed to the solvent.

The new data confirm our general hypothesis in that a proper insertion of the side chain of residue 151 in the pocket (R) promotes a normal channel responds to cAMP. If the chain (D,K) is not inserted in the pocket (solvated) the channel is not measurable due to altered folding/trafficking.

3) Most of the mutations in the first interaction site (Figure 2) result in channels that don't make it to the membrane, however, the I176A and L250A mutants both do something interesting to gating. Is the voltage dependence of these channels lost or just shifted? What do the IVs look like? Did you try to hold more depolarized to close the channels?

We have performed experiments on I176A extending the voltage range to more depolarized values. The IV curves from three different cells, Author response image 1, indicate that the mutant has not lost the voltage –dependency but the relative portion of channels opening in a voltage-independent manner is increased.

**Author response image 1. respfig1:** Patch clamp analysis of HCN2 I176A mutant. (**A**) Representative whole-cell currents of HCN2 wt and I176A channels recorded from +50 to -120 mV with -15mV increment. A wt trace with a low current amplitude similar to those of I176A was selected for this comparison. Scale bar is 100 pA x 500 ms (**B**) Current-voltage plot of traces in (**A**). HCN2 wt is shown in black; I176A curves are shown in red.

4) In the second hydrophobic interaction site (Figure 3, Figure 4), you use MD to look at how the local structure changes when the central PHE is mutated and show a disruption of the structure. This is reported as a change in distance between the mutated residue and a nearby TYR. Some clarification about the relationship between the simulation and the data are needed. The three largest changes in the simulation are caused by the mutations to VAL, ALA, and GLU which all perturb the measured distance by the same amount. However, the functional data reports a dramatically different voltage dependence between these three mutants and in the last case no sensitivity to cAMP. It is unclear how the MD simulation relates and enhances these data.

We have calculated the correlation between the shifts in V_1/2_ in Figure 4B (and Supplementary file 1A) and the distances in Figure 3C and got a spearman correlation coefficient of 0.83 with a p-value of 0.04. Nonetheless, we agree with the reviewers that this correlation is not so obvious, and it is difficult to relate the plot to the e-phys results. Probably the solvation plot is much clearer to interpret in the view of the experimental data. This plot further enhances the idea that the change in V_1/2_ and the loss of cAMP in F151E is a direct effect of the disruption of the contact within the hydrophobic core.

We decided to make this message clearer in the new Figure 3 by removing Panel 3c and substituting Panel 3B. Panel 3B now illustrates a 50 ns long MD simulation showing that the wt F151 is stably inserted in the pocket while the F151E mutant moves permanently outside of the pocket. We have further improved the message of Panel 3D by simplifying it. The new Panel 3D shows the distance from the first hydrated shell only.

The text in the results has been changed accordingly subsection “Second hydrophobic pocket”).

5) In a number of places in the hydrophobic site (I134A versus I176A or F109 versus F151) there is a substantial difference between your results in HCN1 versus HCN2, whereas the C-linker HCND interaction results are quite consistent. Some comment on the ubiquity of this mechanism in the HCN family might be warranted.

It is true, consistency among subunits is maximal at the C linker level, while at the TM domain level differences emerge. HCNs isoforms differ substantially in the extent and kinetics of cAMP response, a difference which is difficult to relate to their highly conserved cytosolic domains. From our data it emerges that the differences between isoforms might be related to the TM domains or, better, to the way in which the cytosolic domain connects to the TM domains.

We have added a sentence in the Discussion section.

6) A little more discussion of linear response theory and what the modeling can tell you and its limitations might be helpful. We are unfamiliar with this approach and a broad audience might be as well. If you break the putative bonds/salt bridges does the model tell you that the force is not transmitted as well?

We have added some explanation on LRT in the text, subsection “C-linker contacts”.

As for the specific question of what happens if we break a putative salt bridge in the LRT model, it is possible to mathematically “switch off” critical contacts and follow the impact on the mechanical connections in the protein. In the present case this method is unfortunately not sufficiently sensitive because the “switched off” contacts are surrounded by a network of other bonds, which interfere with this analysis. We are currently working on improving the sensitivity of the method by including information on chemical interactions. But this is too premature for including in this paper.